# Factors associated with calendar literacy and last menstrual period (LMP) recall: a prospective programmatic implication to maternal health in Bangladesh

Bidhan Krishna Sarker [ID] ,[1] Musfikur Rahman,[1] Tanjina Rahman,[1] Tawhidur Rahman,[1] Fariya Rahman,[1] Jubaida Jahan Khalil,[1] Mehedi Hasan,[1] Sadia Nishat Mahfuz,[2] Faisal Ahmmed,[3] Muhammad Salim Miah,[4] Anisuddin Ahmed,[1] Dipak Mitra,[5] Malay Kanti Mridha [ID] ,[6] Anisur Rahman [ID] [1]

► Prepublication history and supplemental material for this paper is available online. To view these files, please visit the journal online (http://dx.doi.org/10.1136/bmjopen-2020-036994).

For numbered affiliations see end of article.

**Correspondence to**
Mr Bidhan Krishna Sarker; bidhan@icddrb.org

## ABSTRACT

**Objective** To explore the prevalence and determinants of calendar literacy and last menstrual period (LMP) recall among women in Bangladesh.

**Design** Cross-sectional survey.

**Settings** Two rural subdistricts and one urban area from three Northern districts of Bangladesh.

**Participants** We interviewed 2731 women who had a live birth in the last 1 year.

**Primary and secondary outcome measures** The primary outcome variable was LMP recall and the secondary outcome was calendar literacy.

**Results** The majority of participants (65%) correctly mentioned the current date according to the English calendar while 12% mentioned according to the Bengali calendar. During the interview sessions, we used three different calendars: Bengali, English and Hijri to assess calendar literacy. We asked women to mark the current date using the calendar on the day of the interview. Almost 61% women marked the English calendar, 16% marked the Bengali calendar and 4% marked the Hijri calendar correctly. Sixty-three per cent women were found as calendar literate who marked any of the calendars. Among the participants, 58% had calendars available at their home and only 10% of women used calendars to track their LMPs. Overall, 53% women were able to recall their recent LMP. Among the calendar literate, 60% could recall their LMPs. Factors found associated with recalling LMP were: completed eight or more years of schooling (adj.OR 1.39), primigravida (adj. OR 1.88), the richest wealth quintile (adj.OR 1.55) and calendar literacy (adj.OR 1.59).

**Conclusions** Despite having reasonable calendar literacy and availability, the use of calendars for tracking LMP found very low. Calendar literacy and sociodemographic characteristics were found as the key factors associated with LMP recall. Maternal, neonatal and child health programmes in low-resource settings can promote a simple tool like calendar and target the communities where ultrasound is not available to ensure accurate LMP recall for early pregnancy registration and timely antenatal care coverage.

## Strengths and limitations of this study

► This is the first large-scale study of its kind in Bangladesh, reporting the prevalence of last menstrual period recall in urban and rural settings.

► The study relied on the self-reported data, but the inbuilt quality assurance system minimised the errors and improved the study results' validity.

► We took into account both government and non-government health service catchment areas maintaining similar population coverage and statistical rigour.

► We conducted this study in the Northern part of Bangladesh considering higher antenatal care (ANC) coverage, moderate literacy rate and low human resource gaps; hence, the findings are subject to compare contextually.

► Several important factors such as lack of awareness about the importance of gestational age and irregular menstrual cycle were not explored which may influence the risk estimates in this study.

## INTRODUCTION

Every day about 830 women die from pregnancy or childbirth-related complications around the world. Most of these deaths occur in low-resource settings and many of them are preventable.[1] The South Asia region alone accounts for approximately one-third of the global maternal and child deaths annually.[2] Globally, the new target set by the Sustainable Development Goals for maternal health is to reduce the mortality ratio less than 70 per 100 000 live births by 2030.[1] According to the recent national data, the maternal mortality ratio in Bangladesh remains unchanged for the last decade (196 deaths per 100 000 live births) and a considerable effort is needed to achieve the Sustainable Development Goals.[3 4]

Pregnancy is an important part of women's lives and women should have awareness regarding pregnancy care.[5–8] Reliable information about the gestational age is crucial to ensure optimal pregnancy care.[9 10] To detect the gestational age, recall of last menstrual period (LMP) plays an important role. Recalling of LMP is also known as the 'calendar method' for predicting the next ovulation date. The calendar method is based on the recognition of cycles in the menstrual period and fertility, in which women record their menstrual cycles.[6] The WHO also recommends recalling LMP for detecting gestational age because it is a simple and low-cost method.[11 12] The LMP method has been widely accepted and used in many low and middle-income countries for calculating the gestational age and also a suggested recall method where ultrasound facilities are not readily available.[10 13–15]

Although LMP is used in low-resource settings, including Bangladesh, it is well known that the methods have inherent errors in identifying the LMP date correctly. In most cases, inaccurate LMP recall overestimates the gestational age.[12 16] Furthermore, studies conducted in both developed and developing countries have reported that 15%–45% of women cannot recall their LMPs accurately.[12 16–18] The study has also pointed out that the validity of gestational age calculation with LMP depends on better maternal recall of dates and cycle characteristics, which is somehow related to literacy.[14]

Accurate LMP recall helps to screen for any chance of pre-eclampsia, fetal growth restriction and premature birth outcome.[15 16 19–21] Accurate LMP recall also enables detecting pregnancy sooner and facilitates early initiation of pregnancy care. Initiation of antenatal care at the right time is crucial because the first 8 weeks of gestational age are very vulnerable for mothers and fetuses.[22]

Calendar literacy can be an integral part of recalling LMP accurately. We carried out this study recognising the knowledge gap and understanding the need to know the underneath factors that expedite the calendar literacy rate and LMP recall. Therefore, this paper aims to explore the prevalence and determinants of calendar literacy and LMP recall in urban and rural areas from three Northern districts in Bangladesh.

## MATERIALS AND METHODS
### Study design and settings
We conducted this cross-sectional study in rural and urban areas in the Northern districts from Bangladesh. The rural sites were located in the Chirirbandar and the Saidpur subdistricts under the districts of Dinajpur and Nilphamary, respectively. The urban site was located in the city corporation of Rajshahi district. According to the population and housing census (2011), Chirirbandar consisted of 142 villages under 12 unions.[23] The total population of Chirirbandar was 292 500 of which 145 881 were women. The average size of the population of each union was 24 375. Saidpur subdistrict consisted of one municipality and five unions with 39 villages.[24] The

total population of Saidpur was 264 461 of which 130 724 were women. The average size of the population of each union was 27 471. Rajshahi City Corporation consisted of 30 wards.[25] The total population of the Rajshahi City Corporation was 449 756 of which 216 782 were women. The average size of the population of each ward was 14 992. According to the census data, the female literacy rate was 42% in Dinajpur and 39% for both Nilphamari and Rajshahi districts.[26] Healthcare utilisation data suggested that at least one antenatal care (ANC) coverage in Dinajpur, Nilphamari and Rajshahi was 75%, 96% and 80%, respectively.[27] Another data showed a medium vacancy (21%–30%) in terms of healthcare personnel in Dinajpur, Nilphamari and Rajshahi districts.[28] We considered moderate literacy rate, high ANC coverage and moderate vacancy for human resources to select the study areas.

We considered the area served by the government and non-government community-based health workers as a unit or cluster in rural and urban sites maintaining similar population coverage. The women from each cluster who met the inclusion criteria were invited to participate in the survey. The followings were the inclusion criteria for the study participants' enrolment: (1) had a live birth outcome in the last 1 year, (2) past 28 or more days after the last delivery, (3) could hear, see and speak, (4) had permanent residence in the study area and (5) provided informed consent to interview within 2 days of the first contact. The data collection period was from August to November 2016. In total, 2731 women participated in the study.

### Sampling and study participants
We applied two-stage cluster sampling to select study participants in the study area. In the first stage, we randomly selected 1 subdistrict from 2 districts each and 10 wards (lowest administrative unit of city area) from Rajshahi City Corporation. Then, we randomly selected 6 clusters out of 12 in the Chirirbandar subdistrict, 6 clusters out of 12 clusters in the Saidpur subdistrict and 6 clusters from 10 clusters in Rajshahi. We used a conventional statistical method to calculate the sample size. Based on the selected process indicators and assuming the improvement rate after project implementation, we calculated the sample size for the survey. As this cross-sectional study was conducted to get a baseline status in designing an intervention focusing on the improvement of maternal and child health, we considered the proportions at baseline and end line to estimate the sample size. The sample size was calculated using the formula proposed by Hayes and Bennett.[29] Considering the proportion of taking any ANC by the first trimester $p_0$=0.41 (baseline), $p_1$=0.57 (end line) and K=0.15 (coefficient of variation between clusters), we needed the highest number of 150 mothers in each cluster, including 5% non-response rate. Considering the proportion of taking at least one ANC from medically trained provider $p_0$=0.49 and $p_1$=0.66 and K=0.15, we needed the highest number of 6 clusters per area to assess the change in the indicators those were

least known at baseline. For each study area, the sample size was calculated 900; the total required sample size was 2700 (150×6×3).

We recruited study participants through the expanded programme on immunisation method.[30 31] At first, we selected the starting point to start data collection in the selected clusters. We consulted with the community people to determine the mid-point of each cluster. We used the bottle spinning method to ensure randomisation. We spun the bottle at the midpoint of each cluster to pick a random direction to start searching study participants. Afterwards, we visited every household on the next door basis as per bottle indicated direction, and interviewers identified eligible participants for interviewing. Data collection continued until the cluster's sample size was met. Though the required sample size was 150 in each cluster but in some cases, the data collection team interviewed one or two more women to cover the entire cluster. Finally, we completed 2731 interviews from the three study areas. If any eligible woman was absent during the household visit, then data collectors tried at least two more times to interview her.

## Data collection
### Survey tool development
We used a semistructured questionnaire to collect data through face-to-face interviews with women. A team of experienced investigators from icddr,b was responsible for developing this survey tool. Based on the research objectives and the expected outcome variables of the study, the research team considered the relevant indicators to design the tool. As part of designing the tool, investigators did the desk review of the relevant literature such as—Bangladesh Demographic and Health Survey, Bangladesh Maternal Mortality and Healthcare Survey and other similar studies. After formulating the tool, we carried out a pretest to check the design of the questionnaire how it works in practice. After addressing all the problems related to the survey tool, the data collection team started collecting data. In the survey tool, we included the basic individual and household characteristics of the study participants (age, religion, years of schooling, employment status, income, household asset and place of residence); reproductive and obstetric information of the participants (age at first marriage, gravida, para, living children, pregnancy history, LMP date), knowledge and practice of calendar literacy (able to mention correctly Bengali and English day, month, year and date which is related to their knowledge; able to mark Bengali/English/Hijri date in the calendar which is related to their practice).

### Staff recruitment and training
We involved a team of efficient and experienced field research assistants for data collection. We recruited field research assistants who had a minimum Bachelor's degree with at least 2 years of previous experience in the relevant field especially in similar positions in large-scale surveys. After recruitment, we trained the interviewers on the survey tool to collect data. During training sessions, the key persons from the research team discussed the objectives of the study so that the data collection team can easily avoid any confusion. Furthermore, we developed a training manual for the interviewers that included interviewing techniques and specific instructions for each of the data collection questionnaire and checklist.

## Data management and quality assurance
A project research physician and a research investigator coordinated the data collection team. Furthermore, we divided the data collection team into three groups. Each group had four data collectors and a group leader. The group leaders were responsible for ensuring quality data regularly. They checked the completeness of every interview on the spot after the data collected through the day-to-day monitoring. Again, as per the monitoring plan, the group leaders and the team coordinators reinterviewed almost one-third of the total interviewees to check the accuracy and the validity of data. We also checked relevant documents (such as calendar) during our data collection to minimise the errors. Expert programmers of the Maternal and Child Health Division of icddr,b designed a database template maintaining skipping options strictly and logically to avoid inconsistency of data. The programming team used Dot net (V.10) software for data template design as appropriate. An expert data management team was engaged in entering all the data through an online database. The data management team simultaneously entered both the precoded and the postcoded data. For postcoding data, the research team was closely involved with the data management team.

## Ethics
icddr,b has rigorous processes to meet research integrity and ethics, which is called the Institutional Review Board (IRB). icddr,b's IRB consists of four mandatory committees involving internal and external topic experts: Research Review Committee (RRC), Ethical Review Committee (ERC), Animal Experimentation Ethics Committee and Programme Coordination Committee. icddr,b attaches great importance to ensuring that all research protocols conducted by its scientists are scientifically sound and meet international ethical standards. To achieve this, in addition, to review at the centre level and by at least two external reviewers, all research protocols are subject to review by at least two of three committees: RRC, ERC (protocols involving human subjects) and the Animal Experimentation Ethics Committee (protocols involving animals).

RRC has the authority to approve, suggest modifications to or disapprove of any proposed research protocol. Any change to an approved research protocol also needs the approval of the RRC. No research protocol can be implemented or proceed for consideration by the ERC or the Animal Experimentation Ethics Committee without the RRC's approval.

The ERC's approval is required for all protocols involving human subjects. It is an independent body committed to protecting the rights of the people who participate in research protocols conducted under the auspices of icddr,b. A subcommittee of the ERC, the Data Safety Monitoring Board, periodically reviews and evaluates the accumulated study and makes recommendations to the ERC concerning the continuation, modification or termination of the study.

We acquired both RRC and ERC approval from the IRB of icddr,b before we start collecting data. During the data collection period, we acquired written informed consent from all the participants after ensuring that none of them is exposed to minimal risk while participating in this study. Since all our participants were already married by then, we did not need to take consent for the minors from their guardians or parents as per IRB requirements.

### Variables and measures

All the information given by the mothers were self-reported. The primary outcome variable was LMP recall and the secondary outcome was calendar literacy. The rest of the variables were considered as independent variables. The description of all variables is given below—

| List of variables | Explanation |
| --- | --- |
| Last menstrual period (LMP) recall (primary outcome variable) | We considered the first day of LMP. We asked the women about the LMP date of their menstrual cycle prior to the interview date. |
| Calendar literacy (secondary outcome variable) | We asked study participants to mark the current date from the three different types of calendars (Bengali, Hijri and English). If the study participants marked it correctly, we coded '1' for the correct answer and '0' for the wrong answer. We considered women as calendar literate if they could accurately mark the current date using any of the provided calendars. |
| Mentioned the current day, month, year and date | We asked women whether they could tell the name of the present day, month, year and the date (the current day of interview) |
| Age in years | We considered completed years of age and categorised into the following ranges ≤19, 20–29, ≥30 years of age |
| Years of schooling | We considered completed years of schooling and categorised into <5, 5–7 and ≥8 years of schooling. |
| Place of residence | We considered interviewing the women with permanent residence at the respective study sites. We categorised place of residence into rural and urban areas. |
| Employment status of the participants | It consisted of multiple categories such as unemployed, service, business, handicraft, agriculture, farm/fishing and day labour. |
| Gravida | The total number of all pregnancies reported by a woman that she had in her lifetime. |
| Availability of calendar at home | The calendar was available at home during the interview. |
| Purpose of using the calendar | We asked the study participants about the purposes of using a calendar. Multiple responses were included such as—calculate days and dates, track the dates of the menstrual period, remember the special/festival day (Hat day {weekly shopping day at village}/Eid/Puja/Selling certain goods), remember the date of the tuition fee of children, remember the loan instalment day, child's date of birth and track lunar date. |
| Wealth Index | The Wealth Index was calculated using easy-to-collect data on a household's ownership of selected assets including primary source of drinking water, boiled water prior to drinking, the water source for cooking and handwashing, toilet facility, cooking fuel, main construction material of the roof, the main material of the exterior walls, the main material of the floor, rooms used for sleeping, ownership of farm animals, own household, own agricultural land. Besides these, ownership of durable goods such as—radio, television, mobile phone, refrigerator, shallow machine, computer; ownership of transport such as—bus, truck, rickshaw, van, motorcycle. The Wealth Index was a categorical variable that categorised into poorer, poor, middle, rich and richest. |

### Statistical analysis

We performed a quantitative analysis using the statistical software package STATA, V.13. We used both descriptive and analytical approaches. The study participants' characteristics, including their calendar literacy and the capability of recall LMPs, were presented by mean, median and proportion. To examine the association between outcomes and available covariates, we performed a logistic regression analysis. The potential confounders such as maternal age, place of residence, years of schooling, employment status, and economic condition by asset quintiles were adjusted in the regression analysis. The results were presented by OR with 95% CI. All statistically significant results were reported at p<0.05.

We generated Wealth Scores by using the principal component analysis method.[32 33] The Wealth Index is a

**Table 1** Sociodemographic characteristics of the survey participants

| Traits | Total n=2731 (%) |
|---|---|
| Age in years | |
| ≤19 | 609 (22.30) |
| 20–29 | 1693 (61.99) |
| ≥30 | 429 (15.71) |
| Completed years of schooling | |
| <5 | 541 (19.81) |
| 5–7 | 796 (29.15) |
| ≥8 | 1394 (51.04) |
| Residence | |
| Urban | 915 (33.50) |
| Rural | 1816 (66.50) |
| Employment status of the participant | |
| Unemployed | 2583 (94.58) |
| Employed* | 148 (5.42) |
| Gravida | |
| 1 | 1038 (38.01) |
| ≥2 | 1693 (61.99) |
| Median no. of pregnancies (gravida) | 2 |
| Wealth Index | |
| Poorer | 547 (20.00) |
| Poor | 546 (20.00) |
| Middle | 546 (20.00) |
| Rich | 546 (20.00) |
| Richest | 546 (20.00) |

*Refers to service/business/handicraft/agriculture/farm/fishing, day labour and so on.

composite measure of a household's cumulative living standard.[34] Information on the Wealth Index is based on the data collected through the household questionnaire. Each household was assigned a standardised score for each asset and the collective score differs from one another depending on whether the household owned the assets or not. These scores are summed up by the availability of the household assets and individuals are ranked according to the total score of the household. The resulting asset scores are standardised concerning a standard normal distribution with a mean of 0 and a SD of 1. The sample is then divided into population quintiles—five groups with the same number of individuals in each. These standardised scores are then used to create the breakpoints that defined wealth quintiles as: poorer, poor, middle, rich and richest.

## RESULTS
### Sociodemographic characteristics of the study participants
Table 1 shows that most of the participants (62%) belonged to the age group of 20–29 years, followed by 22% who were below 20 years of age. Overall, 80% of women had completed 5 years of schooling of which 51% had completed eight or more years of schooling. Among the participants, 95% of women were unemployed. Fifty per cent of the participants mentioned that they had experienced two pregnancies (median 2).

### Calendar literacy, use and availability
#### Participants who could mention the day, month and year instantly without seeing the calendar
We asked the participants to mention the name of the current day, month and year (on the day of interview) instantly according to the Bengali and the English calendar. Figure 1 shows almost all the participants (97%) were able to mention the name of the days as per the Bengali calendar, which was almost the same in both rural and urban areas. On the other hand, 45% of participants were able to mention the name of days from the English calendar while participants from urban areas could mention better than their rural counterpart (56% vs 40%). The majority of the urban women (77%) were able to mention the name of the month from the English calendar rather than the name of a month in the Bengali calendar (44%). More rural women (64%) mentioned the name of the months from the Bengali calendar compared with the English calendar (54%). However, the name of the year was mentioned accurately based on the English calendar (urban 85% vs rural 72%) whereas only a few participants could name the year from the Bengali calendar (urban 7% vs rural 6%). In case of remembering date, majority participants both in urban and rural areas (78% and 58%) were able to correctly mention the present date based on the English calendar while only 12% of the total participants mentioned the date from the Bengali calendar.

### Capability and purpose of using a calendar
Table 2 shows that more than half of the women (61%) marked the date correctly from the English calendar whereas 16% marked from the Bengali calendar and only a few of women (4%) marked from the Hijri calendar. About 63% of women could mark correctly from any of these three calendars (Bengali, English and Hijri), which we considered as calendar literacy. More than half of the participants (58%) had a calendar at their home. About 44% of women reported using it for calculating days and dates, whereas only 10% of women used it to track their menstrual cycle.

### Calendar literacy and its associated factors
#### Association between calendar literacy and sociodemographic characteristics
Online supplemental table 1 indicates that the majority participants (67%) from age groups 20–29 were calendar literate. The employed women (81%) had more calendar literacy than the unemployed women (62%). Most of the women (89%) who completed eight or more years of schooling were calendar literate. Ninety per cent of the

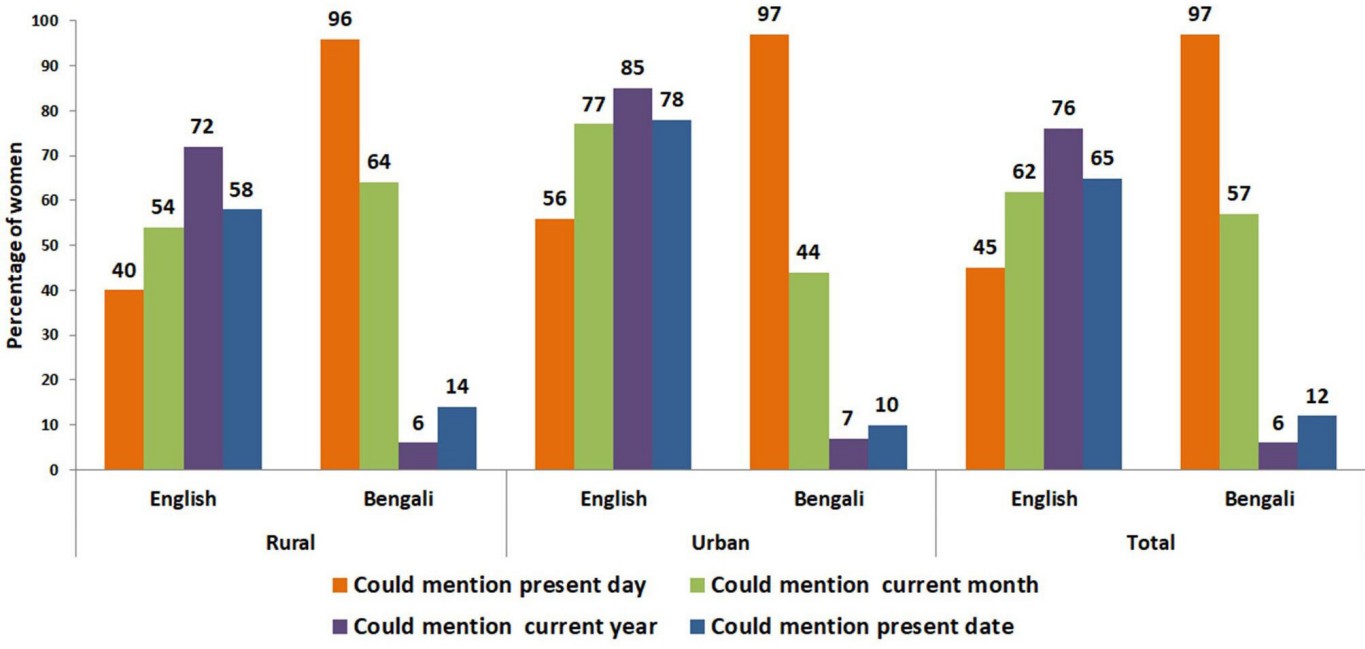

**Figure 1** Women who could mention current date.

women who did not complete 5 years of schooling were calendar illiterate.

### Educational status of calendar literate women

In figure 2, we explored the relationship between calendar literacy and the educational background of

the study participants. Seventy-two per cent of women who completed eight or more years of schooling could mark the correct date in the calendar. However, only a few of the women (3%) who did not complete 5 years of schooling could also mark the calendar correctly.

### Factors associated with calendar literacy

Online supplemental table 2 presents that almost all the indicators were crudely associated with a higher prevalence of calendar literate women. After adjustment, the OR of calendar literate women who completed eight or more years of schooling was 34 times (adj. OR 34.46, 95% CI 23.96 to 49.57) higher than the women who did not

| Table 2 | Capability and purpose of using a calendar |
|---|---|
| **Traits** | **Total n=2731 (%)** |
| Could mark date from the Bengali calendar | 448 (16.40) |
| Could mark date from the English calendar | 1668 (61.08) |
| Could mark date from the Hijri calendar | 114 (4.17) |
| Could mark date from any of these three calendars | 1730 (63.35) |
| Availability of calendar at home | |
| Yes | 1594 (58.37) |
| No | 1137 (41.63) |
| Purpose of using calendar* | |
| To calculate the days and dates | 1204 (44.09) |
| To track the dates of the menstrual period | 268 (9.81) |
| To remember the special/festival day (Hat day/Eid/Puja/Selling certain goods) | 364 (13.33) |
| To remember the date of the tuition fee of children | 85 (3.11) |
| To remember the loan instalment day | 61 (2.23) |
| Others† | 39 (1.43) |

*Refers multiple responses.
†Refers to recall child's date of birth, to calculate the lunar date and so on.

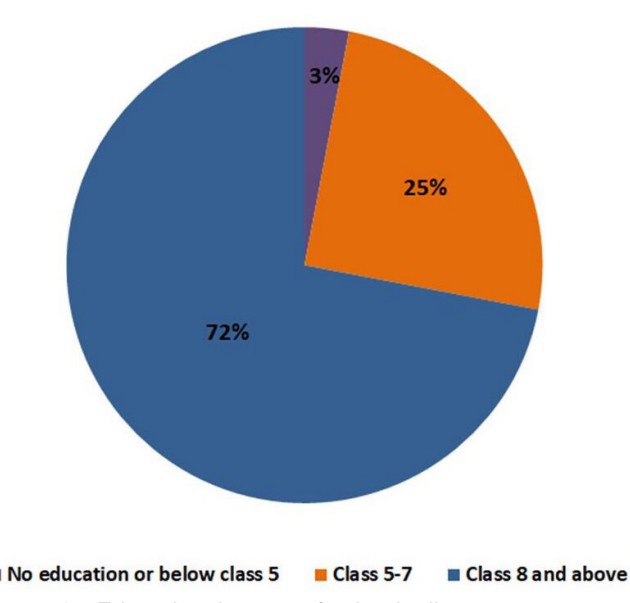

**Figure 2** Educational status of calendar literate women.

complete 5 years of schooling. The women who belonged to the top two wealth quintiles (ie, rich adj. OR 2.75, 95% CI 1.83 to 4.11 and richest adj. OR 3.15, 95% CI 1.88 to 5.29) were almost three times more calendar literate than the poorer women.

## LMP recall

Table 3 presents more than half of the participants (53%) could recall their recent LMP dates. About 59% of women aged ≤19 years recalled their LMP dates. We found a significant statistical (p<0.01) association between the completed years of schooling and the recent LMP recall. The women who had completed eight or more years of schooling, most of them (60%) could recall the LMP date. The women who were calendar literate, about 60% of them could recall their LMPs, and there was a significant (p<0.01) association between the calendar literacy and the recent LMP recall.

## LMP recall and its associated factors

Table 4 present that almost all the indicators were crudely associated with the higher prevalence of recalling LMP. After adjustment, the OR of LMP recall for the women who completed eight or more years of schooling was 1.39 times (adj. OR 1.39, 95% CI 1.05 to 1.84) higher than the women who did not complete 5 years of schooling. Similarly, the richest women were more likely to recall recent LMP than the poorer women (rich: adj. OR 1.46, 95% CI 1.08 to 1.97 and richest: 1.55, 95% CI 1.09 to 2.21). Besides, the women who had primigravida were more likely to recall recent LMP than those who had multigravida (adj. OR 1.88, 95% CI 1.55 to 2.29). The OR of LMP recall for calendar literate women was 1.59 times (adj. OR 1.59, 95% CI 1.28 to 1.98) higher than the calendar illiterate women.

The model adjusted for variables such as woman's age, completed years of schooling, residence, employment status, gravida, Wealth Index, availability of calendar at home, the purpose of using a calendar, tracking LMP and calendar literacy.

## DISCUSSION

This study explored the prevalence of LMP recall and calendar literacy along with the factors associated with these two outcome variables. This study suggested that urban women were more calendar literate than their rural counterparts but we did not find any significant difference between these two groups on the prevalence of recalling recent LMP. This study also found that more than half of the women had calendars available at their home; however, only a few women used the calendar to track their LMPs. As expected, calendar literate women could better recall their LMPs than the calendar illiterate women. Besides, years of schooling, fewer pregnancies and socioeconomic class were the significant associated factors for recalling LMP. Women with eight or more

years of schooling, first time pregnancy and women from the richest wealth quintile could recall their LMP better.

Because of the unavailability of the studies reporting the prevalence of calendar literacy and the factors associated with calendar literacy, and how calendar literacy contributes to LMP recall we could not compare our findings with any study. We could only compare our findings on the prevalence of LMP recall. However, we discussed the use and importance of calendar method or LMP recall in relation to findings from prior studies around the globe including Bangladesh.

Similar to our findings, a study conducted in 2005 in a developed country found that more than half of the women were capable to recall their LMP accurately.[18] A study conducted in Bangladesh in 2016 examined the significance of LMP recall to measure the gestational age by using the home calendar and explored how it differed from the ultrasound result. The study validated the use of the LMP method for estimating gestational age in a rural setting of Bangladesh. The same study also revealed that a mother's higher education level and previous experience of preterm birth are associated with better accuracy of LMP recall.[35] Studies also evidenced that LMP is a reliable way to calculate the gestational age while one study had the precision level of 86%–90% for LMP recall's accuracy and reliability.[13 36]

Several studies prioritised the LMP method to determine the gestational age as well as to assess the preterm birth and fetal growth restriction.[14 37–39] A number of studies used the LMP recall method to identify pregnancy and used paper-based calendar as a tool to get the accurate LMP recall and recruited pregnant women immediately after pregnancy confirmation for their surveillance or interventions.[12–14 18 35–43] Our cross-sectional survey only explored the use of a calendar, calendar literacy, LMP recall and its associated factors in the area where there is no such paper-based calendar intervention. In our country context, women usually remember LMP dates in an unwritten way. This study found that only 10% of women used a calendar for tracking LMP. However, in the modern era, everyone seems to be connected to the digital world, with more and more people using calendars inbuilt in their phone, their email or various other options. Many places like homes and businesses still use at least one printed paper calendar.[44] This study also found that 58% of households owned a printed calendar and 93% owned (according to our household asset calculation) at least one cell phone. The printed calendar provides an easy way to see the date without having to log in and access a tiny screen on the smartphone. So, using a printed calendar to track LMP is still comfortable and relevant for women even in the age of digital technology where an electronic calendar is available on the cell phone.

However, the LMP method has its limitations. LMP can predict gestational age accurately only if the cycle characteristics and the date of onset of the last menstrual bleed can be established.[36 40] Despite the limitations, studies

**Table 3** Association between last menstrual period (LMP) recall and sociodemographic characteristics

| Traits | Recall recent LMP | | P value |
| | Yes 1446 (%) | No 1285 (%) | |
|---|---|---|---|
| **Age in years** | | | |
| ≤19 | 359 (58.95) | 250 (41.05) | <0.01 |
| 20–29 | 899 (53.10) | 794 (46.90) | |
| 30+ | 188 (43.82) | 241 (56.18) | |
| **Completed years of schooling** | | | |
| <5 | 201 (37.15) | 340 (62.85) | <0.01 |
| 5–7 | 406 (51.01) | 390 (48.99) | |
| ≥8 | 839 (60.19) | 555 (39.81) | |
| **Residence** | | | |
| Rural | 948 (52.20) | 868 (47.80) | <0.01 |
| Urban | 498 (54.43) | 417 (45.57) | |
| **Employment status of the participant** | | | |
| Unemployed | 1365 (52.85) | 1218 (47.15) | <0.01 |
| Employed* | 81 (54.73) | 67 (45.27) | |
| **Gravida** | | | |
| 1 | 669 (64.45) | 369 (35.55) | <0.01 |
| ≥2 | 777 (45.89) | 916 (54.11) | |
| **Wealth Index** | | | |
| Poorer | 236 (43.14) | 311 (56.86) | <0.01 |
| Poor | 287 (52.56) | 259 (47.44) | |
| Middle | 294 (53.85) | 252 (46.15) | |
| Rich | 307 (56.23) | 239 (43.77) | |
| Richest | 322 (58.97) | 224 (41.03) | |
| **Availability of calendar at home** | | | |
| Yes | 893 (56.02) | 701 (43.98) | <0.01 |
| No | 553 (48.64) | 584 (51.36) | |
| **Purpose of using calendar†** | | | |
| To calculate the days and dates | 703 (58.39) | 501 (41.61) | 0.04 |
| To track the dates of the menstrual period | 168 (62.69) | 100 (37.31) | |
| To remember the special/festival day (Hat day/ Eid/Puja/Selling certain goods) | 288 (65.45) | 152 (34.55) | |
| To remember the date of the tuition fee of children | 55 (57.89) | 40 (42.11) | |
| To remember the loan instalment day | 33 (54.10) | 28 (45.90) | |
| Others‡ | 22 (56.41) | 17 (43.59) | |
| **Calendar literacy** | | | |
| Literate | 1034 (59.77) | 696 (40.23) | <0.01 |
| Illiterate | 412 (41.16) | 589 (58.84) | |
| **Track LMP date** | | | <0.01 |
| Written | 1394 (53.86) | 1194 (46.14) | |
| Unwritten | 52 (36.36) | 91 (63.64) | |
| **Total** | 1446 (52.95) | 1285 (47.05) | |

*Refers to service/business/handicraft/agriculture/farm/fishing, day labour and so on.
†Refers to multiple responses.
‡Refers to recall child's date of birth, to calculate the lunar date and so on.

**Table 4** Crude and adjusted OR for the factors associated with last menstrual period recall

| Covariates | cOR (95% CI lower to upper) | aOR | (95% CI lower to upper) | P value |
|---|---|---|---|---|
| Constant | | 0.41 | (0.31 to 0.55) | <0.01* |
| The age group of women | | | | |
| ≤19 years | 1.84 (1.43 to 2.36) | 1.10 | (0.81 to 1.49) | 0.53 |
| 20–29 years | 1.45 (1.17 to 1.80) | 1.10 | (0.88 to 1.38) | 0.42 |
| ≥30 years | 1.0 | 1.0 | | |
| Completed years of schooling | | | | |
| <5 | 1.0 | 1.0 | | |
| 5–7 | 1.76 (1.41 to 2.20) | 1.26 | (0.98 to 1.61) | 0.07 |
| ≥8 | 2.56 (2.08 to 3.14) | 1.39 | (1.05 to 1.84) | 0.02* |
| Residence | | | | |
| Rural | 1.0 | 1.0 | | |
| Urban | 1.09 (0.93 to 1.28) | 0.81 | (0.64 to 1.02) | 0.07 |
| Employment status of the participant | | | | |
| Unemployed | 1.0 | 1.0 | | |
| Employed* | 1.08 (0.77 to 1.50) | 0.85 | (0.60 to 1.21) | 0.37 |
| Gravida | | | | |
| 1 | 2.14 (1.82 to 2.51) | 1.88 | (1.55 to 2.29) | <0.01* |
| ≥2 | 1.0 | 1.0 | | |
| Wealth Index | | | | |
| Poorer | 1.0 | 1.0 | | |
| Poor | 1.46 (1.15 to 1.85) | 1.26 | (0.98 to 1.62) | 0.06 |
| Middle | 1.54 (1.21 to 1.95) | 1.28 | (0.99 to 1.66) | 0.06 |
| Rich | 1.69 (1.33 to 2.15) | 1.46 | (1.08 to 1.97) | 0.01* |
| Richest | 1.89 (1.49 to 2.41) | 1.55 | (1.09 to 2.21) | 0.01* |
| Availability of calendar at home | | | | |
| No | 1.0 | 1.0 | | |
| Yes | 1.35 (1.15 to 1.57) | 0.94 | (0.73 to 1.22) | 0.65 |
| Using calendar to calculate days and dates | | | | |
| No | 1.0 | 1.0 | | |
| Yes | 1.48 (1.27 to 1.72) | 0.94 | (0.73 to 1.23) | 0.67 |
| Using a calendar to track the dates of the menstrual period | | | | |
| No | 1.0 | 1.0 | | |
| Yes | 1.58 (1.22 to 2.06) | 1.25 | (0.94 to 1.65) | 0.12 |
| Calendar literacy | | | | |
| Literate | 2.12 (1.81 to 2.49) | 1.59 | (1.28 to 1.98) | <0.01* |
| Illiterate | 1.0 | 1.0 | | |

Significant p≤0.05.
*Refers to service/business/handicraft/agriculture/farm/fishing, day labour and so on.
aOR, adjusted OR; cOR, crude OR.

conducted in different geographical settings such as Bangladesh and Guatemala have recommended LMP to be the chosen method for determining gestational age in low-resource settings.[12 41 43]

Since our study was conducted in the Northern part of Bangladesh, where ANC coverage is higher, the literacy rate is moderate and human resource gaps are low; hence, the findings are subject to compare contextually. The analysis was done depending on self-reported information without having a strong surveillance system; therefore, the scope of over or under-reporting may exist. Because of the self-reported data, LMP recall may vary.

There may be some other confounding factors such as lack of awareness about the importance of gestational age and irregular menstrual cycle were not explored in this study. In Bangladesh, tracking one's menstrual cycle using a calendar is a concept that is yet to become familiar and practised. If women are calendar literate, they will be able to have a clear idea about their menstrual cycle, monitor, and detect any menstrual abnormality. In addition to these, they will be able to calculate the gestational age properly and identify pregnancy at an earlier stage. Early identification of pregnancy can lead to proper and timely utilisation of ANC. This would prevent complications during pregnancy and delivery for the mothers and the babies, thus to improve birth outcomes. We, therefore, feel a high necessity of further nationwide research on this very topic to improve maternal and child health and smoothing the path to achieve Sustainable Development Goals.

## CONCLUSION AND RECOMMENDATION

Accurate recalling of LMP is crucial for maternal health as it helps to calculate the gestational age accurately. It also helps to detect pregnancy early which can influence early initiation of pregnancy care. We found a significant relationship between calendar literacy and LMP recall. Though the calendar literacy rate and the availability of the calendar at home are reasonable; however, the practice of LMP tracking using a calendar is shallow. Besides calendar literacy, factors like years of schooling, place of residence and socioeconomic status have a significant influence on LMP recall too. Mounting all the relevant evidence, we summed up that calendar literacy is an opportunity to introduce a simple tool like paper-based calendar in any maternal, neonatal and child health programmes at communities. And this would be very useful, especially where ultrasound is not available to ensure accurate LMP recall for early pregnancy registration and timely ANC coverage for improving maternal health outcomes in Bangladesh.

**Author affiliations**
[1]Maternal and Child Health Division, icddr,b, Dhaka, Bangladesh
[2]School of Health Sciences, Western Sydney University, Greater Western Sydney, New South Wales, Australia
[3]Infectious Diseases Division, icddr,b, Dhaka, Bangladesh
[4]Department of Anthropology, Shahjalal University of Science and Technology, Sylhet, Bangladesh
[5]Department of Public Health, North South University, Dhaka, Bangladesh
[6]Centre of Excellence for Non-Communicable Diseases and Nutrition, James P Grant School of Public Health, BRAC University, Dhaka, Bangladesh

**Acknowledgements** This research protocol was funded by the Bill & Melinda Gates Foundation. The awarded BMGF grant number is OPP1146943. icddr,b acknowledges with gratitude the commitment of Bill & Melinda Gates Foundation to its research efforts. The authors are grateful to their study participants for their spontaneous participation and sincere commitment to fulfil the research endeavour. icddr,b is also grateful to the Governments of Bangladesh, Canada, Sweden and the UK for providing core/unrestricted support.

**Contributors** BKS, MR, DM, MKM, AA and AR conceived and designed the experiments. BKS, MR, TawR, JJK, MH and MSM performed the experiments. BKS, TanR, TawR, FA, MH, SNM, AA and AR were involved in analysing data. BKS, MR, TawR, JJK, TanR, MH, FA, MSM and FR contributed to developing the tools. BKS, MR, TanR, TawR, FR, SNM, DM, MKM, AA and AR wrote the paper.

**Funding** This research protocol was funded by the Bill & Melinda Gates Foundation. The awarded BMGF grant number is OPP1146943.

**Competing interests** None declared.

**Patient consent for publication** Not required.

**Provenance and peer review** Not commissioned; externally peer reviewed.

**Data availability statement** Data are available upon reasonable request. Unpublished data are available in accordance with icddr,b's data sharing policy. Requests should be addressed to Ms Armana Ahmend, Head, Research Administration, icddr,b; aahmed@icddrb.org. More information can be found here- https://eur01.safelinks.protection.outlook.com/?url=https%3A%2F%2Fwww.icddrb.org%2Fdmdocuments%2Ficddrb%2520Data%2520Access%2520Policy.pdf&data=02%7C01%7Canne.laterra%40care.org%7C73604d2a11bc4750c21608d75e12da16%7Ce83233b748134ff5893ff60f400bfcba%7C0%7C1%7C637081307376649360&sdata=fLeMIOsL2d%2FnkdoDykAoa9kmPxhn3hnJutjwEk0Scl4%3D&reserved=0

**ORCID iDs**
Bidhan Krishna Sarker http://orcid.org/0000-0003-1479-158X
Malay Kanti Mridha http://orcid.org/0000-0001-9226-457X
Anisur Rahman http://orcid.org/0000-0003-1033-5034

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
