## [Reviewer comments · BMJ Open]

ARTICLE DETAILS

TITLE (PROVISIONAL)	Factors associated with calendar literacy and last menstrual period (LMP) recall: a prospective programmatic implication to maternal health in Bangladesh
AUTHORS	Sarker, Bidhan Krishna; Rahman, Musfikur; Rahman, Tanjina; Rahman, Tawhidur; Rahman, Fariya; Khalil, Jubaida; Hasan, Mehedi; Mahfuz, Sadia; Ahmmed, Faisal; Miah, Muhammad Salim; Ahmed, Anisuddin; Mitra, Dipak; Mridha, Malay Kanti; Rahman, Anisur

VERSION 1 – REVIEW

REVIEWER	MARY AMOAKOH-COLEMAN NMIMR, UNIVERSITY OF GHANA
REVIEW RETURNED	14-Apr-2020

GENERAL COMMENTS	Comment: LMP and Calendar Literacy A very useful topic in the field 1. Generally, the written English is very difficult to follow. Many readers will find it difficult to follow. Action needed: The paper needs extensive proofreading and editing. Methods: 2. Describe the setting a bit more, highlighting features of those 3 rural areas that make this study relevant for the setting Use the STROBE statement for Cross-sectional studies to report the study, especially the Methods section3. How was sample size calculated? How did you settle on 900 women per area? What was the variable of interest used for calculating the sample and what was the value? Describe all that fully4. Measures/ Variables: You have named two primary outcome variables. One should be primary and the other secondary. Decide which one is primary and let the write up and results reflect that adequately. The abstract should also reflect that edit. The other variable becomes your secondary outcome variable. Describe all other variables as independent.5. Page 6, what did you calculate for those who marked current day, month, year and date with respect to Bengali, English and Hijri calendar by themselves?6. Education variable: You use completed years and passing years of schooling interchangeably. The two may mean different things. Please use one consistently, and I recommend you use completed years which is universally understood.
--

	7. Homemaker: what do you mean by homemaker under occupation. 8. Gravida: What do you mean by total number of confirmed pregnancies? Confirmed by what and by whom? 9. Household assets (page7): which household assets did you ask about? They are not listed. 10. Analysis: What method of Principal Component analysis did you do and why? 11. Results (Page 9): paragraph starting with Table 2 – should that paragraph not come earlier under results? 12. Table 3: cOR 95% CI – There are many decimal points making it impossible to see what figures you are showing readers. Use consistent decimal points 13. aOR: Which factors are adjusted for these values. Indicate in text under the Tables 14. P-values: please put p-values in the last column so it is clear they are for the aOR and not cOR. 15. Discussion: The first statement under Discussion is not true as per the aim listed under Introduction 16. Incorporate a section under discussion of Calendar availability and actual use, especially in these days of electronic date systems on phones etc 17. What are the study limitations and strengths?
--	--

REVIEWER	Wouter Bakker Athena Institute VU University, Amsterdam The Netherlands
REVIEW RETURNED	21-Jun-2020

GENERAL COMMENTS	Dear authors, Thanks for submitting this paper on LMP recall which is an important topic in low and middle income countries where the maternal health care depends heavily on this. While I understand the importance of the subject and I am impressed by the large amount of interview you managed to conduct, I have some concerns about the manuscript. Abstract Please rephrase your abstract since a few sentences are unclear, especially the last one of the conclusion. Strengths and limitations Your second point states validation of the data by re-interviewing, but I did not get this clearly from your methods. Your only limitations is that the study can't demonstrate a nationwide prevalence. You have a small sentence on this in your
---

discussion, but could you elaborate more on this, why is this part not representative for the country and if so, why was decided to perform the study in this area and with these health centres. Also I think there could be some other limitations named for example the many confounding factors for LMP recall.

Introduction

This part needs quite some attention. Many sentences are not formulated in correct English and the interpunction needs improvement. Please delete all the commas at the end of every sentence and ensure good flow of the paragraphs. Be more concise in stead of general terms as obstetric and paediatric point of view.

In your use of previous literature you switch a lot between national data and then international, in my opinion incomparable data from the US.

Can you elaborate on the calendar method that is widely used? Please also rephrase the last sentence of your introduction.

Materials and methods

I find the methods section quite unclear. Can you elaborate on the tool and (?) questionnaires that were developed for this project? What was the training for data collectors? How were the households selected? You explain the clusters and the random selection but how were the 900 women in that area selected? Why did you decide on the inclusion criteria? Who were the interviewers? What is the institutional research board who gave approval?

The measures section needs better layout, consider using boxes, lists or an image of the questionnaire instead of writing it all in text. Is there a specific reason you decide to use Gravida instead of Para?

Please clarify the use of the different calendars by the participants. Were they supposed to name the dates in all three calendars (English, Bengali and Hijri)?

How was the history of their pregnancies collected?

How was the economic status measured, by what list of household assets? Is this a validated method?

Results

I find it interesting to see that a large majority (80%) had education of more than 5 years but only 63% were capable of using a calendar. Or did you mean using calendars in all three languages. Page 10 paragraph starting with 'Table 2 below shows that..' needs more explanation. After looking at table 2 I think I understand what you mean, but I would suggest changing the phrase any calendar into one or more calendar to make it clearer. Just to clarify: does marking a calendar correctly mean a participate is able to mark the current date or any date on the calendar? Or able to name all days months and years?

Table 3

Please make sure the lay out is correct with 95% CI in brackets

Discussion

While your calculated associations are quite strong and you are able to draw good conclusions out of these, most conclusions are not new. I am surprised by the high proportion of women not able to recall their LMP, since you report other studies which

	recommend using this reliable method. How is care arranged at the moment if almost half of the women can't recall their LMP. Furthermore, still a high proportion of women who is calendar literate is unable to recall their LMP, so there might be other factors involved (education about importance of gestational age?) Can you explain why other studies advise LMP recall as a reliable method whilst your study finds quite a low LMP recall result? The reference [25] of a large multi centre trial in the US is in my opinion not suitable for your situation and also consists of a trial after misoprostol and mifepristone. The article is not accessible online but in the abstract there is no mention of LMP recall. I think there might be better papers into LMP recall in settings more similar to your study setting. Please have another look at the flow of your discussion. Some parts about the use of LMP recall belong more to introduction and there is some repetitive element here. Supplementary table 2: Please include N. How did you calculate the p values for the supplementary tables 1 and 2? Because I don't get a significant value on every variable, for example in table 2 LMP recall rural vs urban does not seem to be a significant difference. Can you clarify what tests and calculations were used? Overall, I think this manuscript is promising but needs a very thorough revision before resubmission. Please make sure it is checked for correct English before submission and make sure all interpunction, tables and other layout is correct to make it readable and understandable. I am happy to review the manuscript again after these improvements.
--	--

VERSION 1 – AUTHOR RESPONSE

Reviewer(s)' Comments to Author:

Reviewer: 1

Reviewer Name: MARY AMOAKOH-COLEMAN

Institution and Country: NMIMR, UNIVERSITY OF GHANA Please state any competing interests or state 'None declared': NONE DECLARED

Comment: Generally, the written English is very difficult to follow. Many readers will find it difficult to follow. Action needed: The paper needs extensive proofreading and editing.

Response: We appreciate your valuable feedback. We have checked thoroughly for settling the grammatical and linguistic errors in our revised paper to make it easier for the readers.

Methods:

Comment: Describe the setting a bit more, highlighting features of those 3 rural areas that make this study relevant for the setting, Use the STROBE statement for Cross-sectional studies to report the study, especially the Methods section

Response: Thank you for your comment. we have added the features of 3 study areas and put the justification for selecting the study area. Please see the study design and setting section

under methodology at the page#7-8. We have maintained the STROBE checklist for cross-sectional studies throughout the paper.

Comment: How was sample size calculated? How did you settle on 900 women per area? What was the variable of interest used for calculating the sample and what was the value? Describe all that fully
Response: We appreciate your valuable feedback. We have added all the detailed information about sample size calculation, the indicators and the value used for calculating sample size in the “sampling and study participant” section under methodology at page#8-9.

Comment: Measures/ Variables: You have named two primary outcome variables. One should be primary and the other secondary. Decide which one is primary and let the write up and results reflect that adequately. The abstract should also reflect that edit. The other variable becomes your secondary outcome variable. Describe all other variables as independent.

Response: Thanks for your constructive feedback. We have revised the outcome variables where primary outcome is LMP recall and secondary outcome is calendar literacy. Rest of the variables are considered as independent variables. It has been reflected throughout the paper including abstract.

Comment: Page 6, what did you calculate for those who marked current day, month, year and date with respect to Bengali, English and Hijri calendar by themselves?

Response: Thank you again for your thorough review and insightful comment. We provided study participants with 3 calendars (Bengali, English and Hijri) and asked them to marked the current date on the calendar on the day of interview (which means day, month and year were correctly marked). We used those data to calculate calendar literacy rate. By calendar literacy, we considered if a woman could correctly mark the current date using any of these calendar. We did not consider them who could mention the name of the day, month and year rather we considered who could mark the current date in practice on the provided calendar properly. We have revised the measures section accordingly. Please see the measures section at page#12-13.

Comment: Education variable: You use completed years and passing years of schooling interchangeably. The two may mean different things. Please use one consistently, and I recommend you use completed years which is universally understood.

Response: Thank you for your suggestion. We have changed it uniformly to completed years of schooling throughout the paper.

Comment: Homemaker: what do you mean by homemaker under occupation.

Response: Thanks for your query. By homemaker we have referred, the women who were housewives and were not involved in any formal employment. However, we have replaced homemaker by unemployed.

Comment: Gravida: What do you mean by total number of confirmed pregnancies? Confirmed by what and by whom?

Response: Thanks again for your question. As universally recognized definition for gravida, we also followed the number of all pregnancies reported by a woman had in her lifetime (till the interview date). We collected the pregnancy history of the participants but we did not collect how the participants confirmed their pregnancies and who confirmed except the recent pregnancy. The recent pregnancies were confirmed by either participants themselves or health workers or relatives. The recent pregnancies were confirmed by pregnancy strip (77.6%), urine test in lab (5.5%), ultra-sonogram (8.9%) and blood test (0.4%).

Comment: Household assets (page7): which household assets did you ask about? They are not listed.

Response: Thank you again for your insightful comment. We collected basic household characteristics of the study participants. The household assets we asked about were main source of drinking water, boiled water prior to drink, water source for cooking and hand washing, toilet facility, cooking fuel, main construction material of the roof, main material of the exterior walls, main material of the floor, rooms used for sleeping, ownership of farm animals, own household, own agricultural land. Besides these, ownership of durable goods such as- radio, television, mobile phone, refrigerator, shallow machine, computer; ownership of transport such as- bus, truck, rickshaw, van, motorcycle, etc. Please see the measures section under methodology at page #12-13.

Comment: Analysis: What method of Principal Component analysis did you do and why?

Response: Thank you again for your valuable questions. We followed standard steps of principal component analysis for measuring the wealth index of the households. We have added the details in the statistical analysis section under methodology at page #14.

Comment: Results (Page 9): paragraph starting with Table 2 – should that paragraph not come earlier under results?

Response: Thank you for your observation. We have revised it accordingly. Please see the revision at page#17.

Comment: Table 3: cOR 95% CI – There are many decimal points making it impossible to see what figures you are showing readers. Use consistent decimal points

Response: We agree with you on this point and changed it accordingly throughout the paper.

Comment: aOR: Which factors are adjusted for these values. Indicate in text under the Tables

Response: We have inserted the factors under the tables as per your suggestion.

Comment: P-values: please put p-values in the last column so it is clear they are for the aOR and not cOR.

Response: We have put the p-values in the last column of table to avoid the confusion.

Comment: Discussion: The first statement under Discussion is not true as per the aim listed under Introduction

Response: Thanks for your feedback. We have revised the statement accordingly. Please see the first line of the discussion at page#25.

Comment: Incorporate a section under discussion of Calendar availability and actual use, especially in these days of electronic date systems on phones etc.

Response: Thanks again for your suggestion. As per your suggestion, we have added a section on “calendar availability and actual use, especially in these days of electronic date systems on phones” under discussion at page #27.

Comment: What are the study limitations and strengths?

Response: Thank you for your question. As per submission guideline by BMJ Open, we placed our study limitations and strengths immediately after abstract at page#4. Besides, we have also added the methodological limitations in the discussion section. Please see the last paragraph of the discussion section at page#27.

Reviewer: 2

Reviewer Name: Wouter Bakker

Institution and Country:

Athena Institute

VU University, Amsterdam

The Netherlands

Please state any competing interests or state 'None declared': None declared

Abstract

Comment: Please rephrase your abstract since a few sentences are unclear, especially the last one of the conclusion.

Response: Thank you for your valuable feedback. We have revised our abstract to make it more clear as per BMJ Open abstract format. Please see the abstract at page#2-3.

Strengths and limitations

Comment: Your second point states validation of the data by re-interviewing, but I did not get this clearly from your methods.

Your only limitations is that the study can't demonstrate a nationwide prevalence. You have a small sentence on this in your discussion, but could you elaborate more on this, why is this part not representative for the country and if so, why was decided to perform the study in this area and with these health centres. Also I think there could be some other limitations named for example the many confounding factors for LMP recall.

Response: Thanks for your minute observations. We have inserted re-interview related information in the data management and quality assurance section under methodology. Please see the page at #10-11.

We have tried to elaborate our limitations and added confounding factors for LMP recall in both "Strength and limitation of the study" section and in the discussion. Please see the changes at page #4. We have also added the justification for selecting the study areas in the study design and setting section at page #7-8.

Introduction

Comment: This part needs quite some attention. Many sentences are not formulated in correct English and the interpunction needs improvement. Please delete all the commas at the end of every sentence and ensure good flow of the paragraphs. Be more concise in stead of general terms as obstetric and paediatric point of view.

In your use of previous literature you switch a lot between national data and then international, in my opinion incomparable data from the US.

Can you elaborate on the calendar method that is widely used?

Please also rephrase the last sentence of your introduction.

Response: Thanks for your insightful comments. We have given a through look to fix all the inconsistencies including linguistics and grammatical mistakes throughout the introduction. Further, we have revised the flow of the introduction to make it easier to understand for the readers.

Additionally, as per your suggestion we deleted the incomparable data from the US and elaborated the calendar method precisely. Please see the changes at the introduction section at page#5-7.

Materials and methods

I find the methods section quite unclear. Can you elaborate on the tool and (?) questionnaires that were developed for this project? What was the training for data collectors? How were the households selected? You explain the clusters and the random selection but how were the 900 women in that area selected? Why did you decide on the inclusion criteria? Who were the interviewers? What is the institutional research board who gave approval?

Response: Thanks for your minute observations. As per your suggestion, we have included the information about the survey tool, interviewers and their training in the data collection section under methodology at page #9-10. The methods we used to select the household for this study has been added in the sampling and study participants section under methodology at page #8-9.

As per your questions regarding inclusion criteria, we considered the women who had live birth outcome in last one year prior to interview to reduce the recall bias. We also considered past 28 or more days after the delivery for getting post-partum care related information for mother and newborn. We recruited women who had permanent residence in the study area so that it represents the context specific information that is aligned with the availability of local human resources and service coverage. Further, to avoid information gap we selected women who could hear, see and speak. Lastly, to maintain the timeframe of the study, we included women who gave permission to interview within two days.

icddr,b have rigorous processes to meet research integrity and ethics which is called Institutional Review Board (IRB). The details regarding the IRB has been added to the ethics section under methodology at page #11-12.

The measures section needs better layout, consider using boxes, lists or an image of the questionnaire instead of writing it all in text.

Is there a specific reason you decide to use Gravida instead of Para?

Please clarify the use of the different calendars by the participants. Were they supposed to name the dates in all three calendars (English, Bengali and Hijri)?

How was the history of their pregnancies collected?

How was the economic status measured, by what list of household assets? Is this a validated method?

Response: Thanks for your valuable feedback. We have added a box including all variables under measures section instead of writing it all in text. Please see the page at #12-13.

There was no specific reason to select gravida but we found similar trend between gravida and para among the respondent. Our primary outcome of this analysis was LMP recall and we considered gravida instead of para thinking that majority (84%) women were from young age (less than 30 years). We checked with para and there were no significant differences between outcome and independent variables.

We provided them all 3 calendars (Bengali, English and Hijri) and asked them to identify the current date on the calendar on the day of interview (which means day, month and year were correctly marked). But, to determine the calendar literacy, we considered if a woman could correctly mark the current date using any of these calendar.

We collected all the pregnancy history of the participant in her life time (from first to recent) in a tabular format. The questions were asked to know their pregnancy history included- outcome of the pregnancy, date of delivery, sex of child, current status of the child (alive or dead), if alive, the current age of the children, if dead, then date of the death and reasons for death.

We followed standard steps of principal component analysis for measuring the wealth index of the households and this is a validated method. We have added the details in the statistical analysis section under methodology at page #14.

Results

I find it interesting to see that a large majority (80%) had education of more than 5 years but only 63% were capable of using a calendar. Or did you mean using calendars in all three languages.

Page 10 paragraph starting with 'Table 2 below shows that..' needs more explanation. After looking at table 2 I think I understand what you mean, but I would suggest changing the phrase any calendar into one or more calendar to make it clearer.

Just to clarify: does marking a calendar correctly mean a participant is able to mark the current date or any date on the calendar? Or able to name all days months and years?

Response: Thanks again for your insightful comments. As mentioned earlier, we provided study participants all 3 calendars (Bengali, English and Hijri) and asked them to mark the current date on the calendar on the day of interview (which means day, month and year were correctly marked). By calendar literacy (63%), we considered if a woman could correctly mark the current date using any of these calendar.

As per your suggestion we have revised the write up of Table 2 to make it clearer. Please see the change at page #17.

Table 3

Please make sure the lay out is correct with 95% CI in brackets

Response: Thanks for your feedback. We have changed it accordingly throughout the paper along with Table 3.

Discussion

While your calculated associations are quite strong and you are able to draw good conclusions out of these, most conclusions are not new. I am surprised by the high proportion of women not able to recall their LMP, since you report other studies which recommend using this reliable method. How is care arranged at the moment if almost half of the women can't recall their LMP. Furthermore, still a high proportion of women who is calendar literate is unable to recall their LMP, so there might be other factors involved (education about importance of gestational age?) Can you explain why other studies advise LMP recall as a reliable method whilst your study finds quite a low LMP recall result? The reference [25] of a large multi centre trial in the US is in my opinion not suitable for your situation and also consists of a trial after misoprostol and mifepristone. The article is not accessible online but in the abstract there is no mention of LMP recall. I think there might be better papers into LMP recall in settings more similar to your study setting.

Please have another look at the flow of your discussion. Some parts about the use of LMP recall belong more to introduction and there is some repetitive element here.

Response: Thanks for your thorough and insightful comments. Several studies used LMP recall method as reliable method to identify pregnancy especially in resource poor settings where ultrasound is not available. Since LMP recall is poor, the studies utilized paper-based calendar as a tool to get the accurate LMP recall so that they could recruit pregnant women immediately after pregnancy confirmation as per need of their surveillance or interventions. Our cross sectional survey only explored calendar literacy, availability and use of calendar, prevalence of LMP recall and its associated factor in the area where there was no such paper-based calendar intervention in place. We have agreed on this point that there may be some other confounding factors which were not explored in this study. In our country context women usually remember LMP dates through unwritten way. This study found that only 10% women used calendar for tracking LMP.

As per your suggestion, we have dropped the large multi-center trial in the US study [previous reference 25] from our revised paper.

Further, we have revised the discussion section and moved some parts to introduction make it reader friendly. Please see the changes in introduction at page#5-7 and discussion at page#25-27.

Supplementary table 2: Please include N.

How did you calculate the p values for the supplementary tables 1 and 2? Because I don't get a significant value on every variable, for example in table 2 LMP recall rural vs urban does not seem to be a significant difference. Can you clarify what tests and calculations were used?

Response: Thanks for your query. We have now included N. However, we have moved "supplementary Table 2 Association between LMP recall and demographic characteristics" to main document as "Table 3 Association between LMP recall and demographic characteristics" because we only kept results from the analysis based on primary outcome variable (LMP recall) in the main document and the results regarding secondary outcome (Calendar literacy) analysis have been incorporated as supplementary tables. Please see the changes at page #21-22 in main document.

We calculated p-value for supplementary Table 1 and Table 3 in main document from chi-square test. By chi-square test we measured the association between two categorical variables. That is, we measured association between calendar literacy and other covariates as well as women's age, completed years of schooling, residence, employment status, gravida, wealth index, availability of calendar at home, purpose of using calendar in supplementary Table 1. Similarly, we measured association between LMP recall and other covariates as well as women's age, completed years of schooling, residence, employment status, gravida, wealth index, availability of calendar at home, purpose of using calendar, tracking way of LMP, calendar literacy in Table 3 in main document.

Overall, I think this manuscript is promising but needs a very thorough revision before resubmission. Please make sure it is checked for correct English before submission and make sure all interpunction, tables and other layout is correct to make it readable and understandable. I am happy to review the manuscript again after these improvements.

Response: Thank you for your insightful comments and suggestion. We have taken care of your concern like: inter-punctuation, grammatical accuracy, tables and other layout, etc. in our revised version. Indeed, your recommendations have helped us to improve our manuscript. We have taken a through look over the paper and made necessary changes as per your suggestions.

VERSION 2 – REVIEW

REVIEWER	DR. MARY AMOAKOH-COLEMAN NMIMR, UNIVERSITY OF GHANA, LEGON, ACCRA, GHANA
REVIEW RETURNED	03-Aug-2020

GENERAL COMMENTS	 1. Well done, you have substantially revised this paper. 2. The sample size calculation requires more clarity. Why the need for the two sets of proportions? 3. Table 4 etc, there is still inconsistent use of decimal points. Some places 1 others 2. Please align. 4. You need to still do some editing of the text
---

REVIEWER	Wouter Bakker Athena Institute VU University The Netherlands
-----------------	---

REVIEW RETURNED	27-Aug-2020
-------------

GENERAL COMMENTS	Dear authors Thank you for sending us the revised manuscript and response letter. I can clearly see a lot of work has been done since the last manuscript and is has improved greatly. The methods section is deccribed in detail and the results are presented more clearly. My major comment is still the use of English language in the manuscript, although there is improvement compared to the first version. Still, there are many grammatical errors and incomplete sentences which makes part of the manuscript difficult to read. This accounts especially for the abstract and the discussion. The flow of the discussion should also be improved, try to use less paragraphs and summarize more, take the reader in the direction you want to go into. I would suggest to start with your main findings (1st paragraph), compare these to other studies (2nd paragraph), explain the clinical relevance and describe the limitations and advice for further research (3rd and 4th paragraph). Then you could end with an overall conclusion. I think the research is definitely worth publishing and the manuscript shows this was a very thorough and well performed project. I would advice to ask someone to proofread the paper for English spelling and grammar before resubmitting. A few more minor comments:  - Try to avoid repetition, especially in discussion. There are three sentences starting with 'A study in Bangladesh' - Explain all abbreviations you use at first use, only use abbreviations for terms you use more than three times in the manuscript, otherwise just stick with the full term - Avoid using terms as 'etc.' - Be precise in the message you want to send out, avoid long winding sentences Again, I would be happy to review the manuscript again if you decide to resubmit but would strongly recommend to optimize the use of language and with that the readability first.
---

VERSION 2 – AUTHOR RESPONSE

Reviewer(s)' Comments to Author:

Reviewer: 1

Reviewer Name: MARY AMOAKOH-COLEMAN

Institution and Country: NMIMR, UNIVERSITY OF GHANA Please state any competing interests or state 'None declared': NONE DECLARED

Comment 1: Well done, you have substantially revised this paper.

Response: Thank you so much for your compliments.

Comment 2: The sample size calculation requires more clarity. Why the need for the two sets of proportions?

Response: Thank you so much for your comment. This cross-sectional study was conducted to get a baseline status in designing an intervention focusing on the improvement of maternal and child health. Therefore, we considered two sets of proportions to estimate the sample size. We have clarified this issue in the sampling and study participants section on page#6-7.

Comment 3: Table 4 etc, there is still inconsistent use of decimal points. Some places 1 others 2. Please align.

Response: Thanks for your feedback. We have revised it accordingly considering up to 2 decimal points throughout the paper.

Comment 4: You need to still do some editing of the text

Response: We have edited the text throughout the paper.

Reviewer: 2

Reviewer Name: Wouter Bakker

Institution and Country:

Athena Institute

VU University, Amsterdam

The Netherlands

Please state any competing interests or state 'None declared': None declared

Comment: My major comment is still the use of English language in the manuscript, although there is improvement compared to the first version. Still, there are many grammatical errors and incomplete sentences which makes part of the manuscript difficult to read. This accounts especially for the abstract and the discussion.

Response: Thank you for your valuable feedback. We have revisited the paper with a view to improving the English language in the manuscript along with the revision of the abstract and the discussion sections. Please see the revised abstract and discussion sections on page # 2, 19-21.

Comment: The flow of the discussion should also be improved, try to use less paragraphs and summarize more, take the reader in the direction you want to go into. I would suggest to start with your main findings (1st paragraph), compare these to other studies (2nd paragraph), explain the clinical relevance and describe the limitations and advice for further research (3rd and 4th paragraph). Then you could end with an overall conclusion.

Response: Thank you for your useful suggestion. We have considered your suggestion and revised the discussion section accordingly. Please see the revision on page # 19-21.

Comment: I think the research is definitely worth publishing and the manuscript shows this was a very thorough and well performed project. I would advice to ask someone to proofread the paper for English spelling and grammar before resubmitting.

Response: Thanks for your compliments and advice. We have checked the English throughout the manuscript and also took help from a professional copyediting service.

Comment: Try to avoid repetition, especially in discussion. There are three sentences starting with 'A study in Bangladesh'

Response: In our revised discussion section, we have reorganized the sentences to avoid repetition.

Comment: Explain all abbreviations you use at first use, only use abbreviations for terms you use more than three times in the manuscript, otherwise just stick with the full term

Response: We have revised it accordingly throughout the manuscript.

Comment: Avoid using terms as 'etc.'

Response: Thanks for your suggestion. We have avoided it as necessary.

Comment: Be precise in the message you want to send out, avoid long winding sentences

Response: Thanks for your comment. In our revised manuscript, we have made the sentences precise and more understandable to make it readers friendly.

VERSION 3 – REVIEW

REVIEWER	Wouter Bakker Athena Institute VU University Amsterdam The Netherlands
REVIEW RETURNED	12-Oct-2020
GENERAL COMMENTS	Dear authors,

	Thank you for sending the second revision of this manuscript. It has improved tremendously and turned into an understandable, readable manuscript. You have addressed all previous comments and in my opinion the paper is ready to proceed for publication. The methods are now understandable and well defined The discussion is concise / to the point and reads well Your tables are neat and clear Last remark is that you need to explain the abbreviation icddr,b in first use, since for an international public it might not immediately be clear what institute this is.
--	---